# Life Cycle Assessment for Geopolymer Concrete Bricks Using Brown Coal Fly Ash

**Jingxuan Zhang** [1], **Sarah Fernando** [1,2], **David W. Law** [1], **Chamila Gunasekara** [1], **Sujeeva Setunge** [1], **Malindu Sandanayake** [3] **and Guomin Zhang** [1,*]

[1] School of Engineering, RMIT University, Melbourne, VIC 3000, Australia; david.law@rmit.edu.au (D.W.L.); chamila.gunasekara@rmit.edu.au (C.G.); sujeeva.setunge@rmit.edu.au (S.S.)
[2] Faculty of Engineering, University of Peradeniya, Peradeniya 20400, Sri Lanka
[3] College of Sports, Health and Engineering, Victoria University, Footscray, VIC 3011, Australia; malindu.sandanayake@vu.edu.au
[*] Correspondence: kevin.zhang@rmit.edu.au

**Abstract:** Traditionally, the construction industry has predominantly used Portland cement (PC) to manufacture bricks, as it is one of the most-commonly available building materials. However, the employment of waste industrial material for brick production can lead to a significant improvement in terms of sustainability within the construction sector. Geopolymer bricks made from brown coal fly ash, a promising industrial waste by-product, serve as a potential alternative. Conducting a life cycle assessment (LCA), this study thoroughly evaluated the entire manufacturing process's environmental impact, from source material acquisition and transportation to brick manufacturing, distribution, usage, and end-of-life, for brown coal bricks as compared to PC bricks. The LCA of the brown coal bricks revealed that their primary environmental impacts stemmed from the raw material manufacturing and usage, while exhibiting substantial reductions in ozone depletion, water depletion, and metal depletion. These findings highlighted the environmental advantages of the brown coal bricks and their potential to revolutionize sustainable construction practices.

**Keywords:** life cycle assessment; brown coal fly ash; impact assessment; cost–benefit analysis

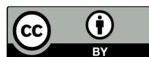

## 1. Introduction

The gravity of climate change and other pressing environmental issues necessitates the prioritization of sustainable solutions. Portland cement (PC) concrete, the most-extensively used building construction material, contributes to an alarming 5–8% of global anthropogenic carbon dioxide emissions [1,2]. The building construction sector has shifted towards sustainable building design, recognizing the significance of the complete project life cycle, from raw material extraction to the disposal stage. The adoption of green strategies throughout a construction project's entire life cycle plays a pivotal role in achieving a building's sustainability. A sustainable construction process should incur minimal environmental impact, not only during the manufacturing and operational stages, but throughout the entire project's life cycle. Roughly 80% of greenhouse gas emissions and energy consumption in the construction sector can be attributed to the operational stages of buildings. In several studies, the implementation of novel technologies, policies, and mitigation measures/technologies has been conducted to reduce GHG emissions during the operational phase [3–5]. Nevertheless, it is equally imperative to curb and lessen the environmental impacts during the initial manufacturing and construction stages.

Brick is a prevalent construction material, and Asia contributes to over two-thirds of global brick production [6]. In Australia, the annual brick production stands at a staggering 1.6 billion [7]. Conventional bricks are typically manufactured from clay with high-temperature kiln firing or PC. Traditional masonry clay bricks are non-eco-friendly due

to the considerable energy consumption [8]. The production of PC consumes a large quantity of raw materials and energy [9,10], which considerably impacts the environment. The production process also emits substantial amounts of greenhouse gases (GHGs) into the atmosphere, exacerbating environmental issues. To alleviate the challenges associated with cement and concrete production, several studies have explored the use of sustainable binders sourced from waste materials. Geopolymer, an alternative to PC binders, holds tremendous promise in the construction sector. Geopolymers are derived by alkali activation of aluminosilicate materials, which could be natural or synthetic materials or industrial by-products [11,12]. To evaluate the environmental impacts of geopolymer binders, life cycle assessment (LCA) is the most-appropriate tool available. It provides consensus framework, terminology, and methodological phases [13,14]. LCA facilitates a comprehensive, quantitative, and interpretive analysis of the environmental impact of a specific service, process, or product over its entire life cycle.

Researchers are exploring innovative ways of producing bricks in a more environmentally friendly manner by repolymerizing alternative waste by-products [15,16]. Brown coal fly ash geopolymer concrete masonry bricks' application presents a sustainable and cost-effective solution for the construction industry. According to estimates, global brown coal production reached 286 Gt in 2016, with Australia being the fourth-largest contributor at 76.5 Gt [17]. In Victoria, Australia, all the brown coal reserves are situated in the La Trobe Valley, primarily at two power plants, Loy Yang and Yallourn, where the brown coal ash is produced from lignite and sub-bituminous coals found in two separate seams [18]. Currently, there are no commercial applications for brown coal ash in the construction industry, and most of it ends up in landfills, leading to environmental contamination. Therefore, utilizing brown coal ash for geopolymer-based bricks would not only minimize the impact of this waste, but also eliminate the need for PC.

The quantification of sustainability factors based on real-life data is crucial to gain awareness in the different environmental impact categories. Early-stage LCA studies are essential to convey the knowledge required to reduce environmental impacts by including the building material manufacturing phases for the entire LCA study. Moreover, to date, most geopolymer brick LCA studies have focused only on the early stages of manufacturing, i.e., cradle-to-gate [19–22], and a limited number of impact categories [23–25]. This study undertook an exhaustive investigation of the LCA of the utilization of waste brown coal ash from the two power plants in the La Trobe Valley, Victoria, in the manufacture of geopolymer bricks, including the twelve major impact categories for the "cradle-to-grave" phases. The study covered twelve major impact categories for the cradle-to-grave phases of the brick's life cycle, offering a detailed analysis of the essential factors that arise from manufacturing brown coal ash geopolymer bricks and the variations due to differences in ash composition. Moreover, this study quantified the environmental benefits of using brown coal ash from landfill sites based on relevant impact categories to enhance the overall understanding of the environmental impact.

This study's objective was fourfold: (1) two distinct types of brown coal fly ash geopolymer bricks' environmental impacts were evaluated and compared with conventional PC bricks; (2) hotspot environmental impact factors were identified and avenues for improvements throughout the entire life cycle suggested; (3) a clear economic analysis of utilizing brown coal fly ash for the production of geopolymer bricks is provided; (4) the impacts based on a comprehensive range of impact categories and real-life data were classified.

Current research on LCA assessment of brown coal ash geopolymer bricks does not include a benefit analysis on the use of performance indicator methods. In order to identify opportunities for the improvement of the environmental and economic performance of brown coal bricks in their production, this study quantified the environmental impacts of two brown coal geopolymer bricks during their "cradle-to-grave" life cycle. Their environmental performance was also compared with that of conventional PC concrete

bricks. The study lays the groundwork for forthcoming research methodologies aimed at maximizing the eco-sustainability of geopolymer bricks made from brown coal fly ash.

## 2. Research Methodology

### 2.1. Research Framework

Utilizing life cycle assessment (LCA), an in-depth study was conducted to scrutinize the environmental impacts and advantages associated with the production of geopolymer concrete bricks composed of brown coal ash sourced from varying power station locations.

Life cycle assessment (LCA) is a systematic approach that utilizes a methodology to gauge the ecological efficacy of products and processes throughout their life cycle, enabling the identification of areas that require refinement [26]. This method has been adopted by researchers, including [27–29]. The LCA procedure described in ISO 14040 [13] was followed to ensure the application of rigorous standards. The ultimate objective of LCA is to quantify and appraise the environmental impact performance of products/processes, facilitating informed decision-making [30]. This framework comprises four distinct, yet interconnected components: goal and scope definition, inventory analysis, impact assessment, and interpretation.

To conduct the impact assessment, the SimaPro (Version 8.2.0) LCA software was employed, which provides a comprehensive and precise analysis. For the purposes of impact assessment, the ReCiPe Mid-Point (Europe H) method (an exceptional feature of SimaPro) was chosen as the best-fitting tool. This sophisticated approach generated a total of eighteen midpoint impact categories, providing a thorough and detailed evaluation [31].

### 2.2. Goal and Scope Definition

The goal of this study was to analyze the environmental impacts and benefits associated with geopolymer concrete bricks manufactured from brown coal ash obtained from two La Trobe Valley power stations in Victoria.

This study also analyzed the economic benefits based on the fly ash's end-of-life storage location. The economic assessment quantified the benefits by linking the environmental impact categories in the comparison unit.

The functional unit selected allowed the normalization of the impacts for the different compressive strengths of the bricks for comparative and contribution analysis in the LCA. A performance indicator approach (unit of functional performance) was adopted for the analysis; thus, this avoided the distinction between the material scale and the structural scale [32].

The functional unit for the process was selected based on the functional performance unit reported by Damineli et al. [32]. This approach compared the environmental impacts associated with variable concrete types and performances (compressive strength) in the LCA study. Hence, a factor termed impact intensity (Equation (1)) was adopted to enable a comparison of the three types of bricks.

$$\text{Impact intensity (i}x) = x/cs \tag{1}$$

where i$x$ is defined as the impact intensity for the "$x$" impact category; $x$ is the total impact derived from the LCA analysis; $cs$ is the compressive strength of the brick.

The economic analysis used the functional unit "1 m$^3$ of brick mixture". The functional unit "1 m$^3$ of brick mix" was employed for the life cycle cost analysis. The "per brick" functional unit was employed for the total cost analysis.

This study considered the "cradle-to-grave" life cycle of products. It consisted of four life cycle stages:

- Raw material extraction/production, which presents the production and preparation of different materials used in the later production stage; those materials included

Na2SiO3, the two brown coal fly ashes, the extraction of aggregates, and the production of the PC. The Na2SiO3, NaOH, brown coal fly ashes, and aggregates were used in the fly ash bricks' production. The aggregates and PC were used to produce the PC concrete.

- The brick production stage represented the transportation of raw materials for the production of the bricks and the production process.
- Distribution and usage represented the transportation of the bricks and the brick wall construction process.
- End-of-life represented the transportation of demolished brick walls and the landfill.

The system boundary of the geopolymer and PC brick wall construction is presented in Figure 1.

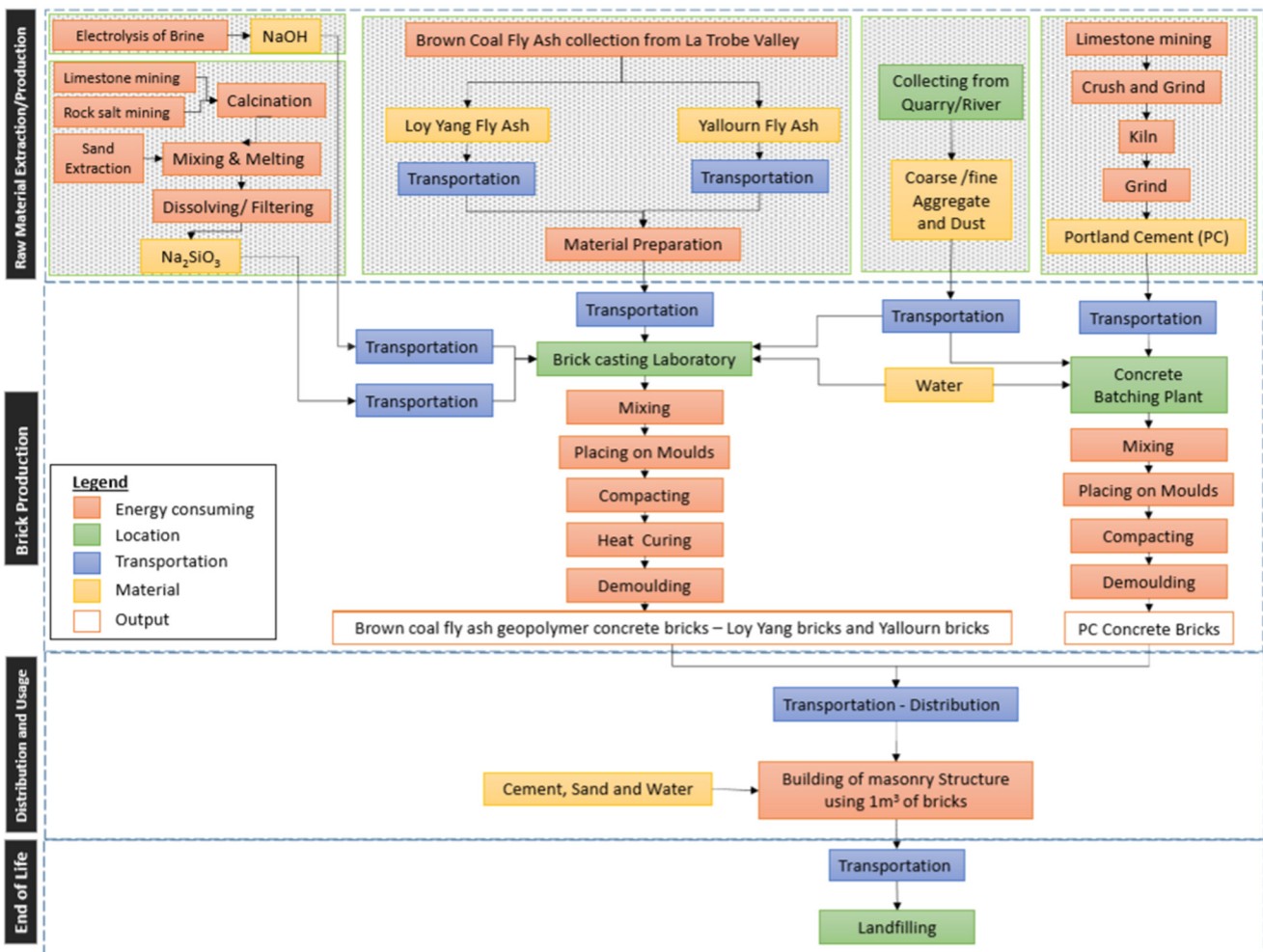

**Figure 1.** System boundary of the life cycle for the concrete brick wall.

### 2.3. Models and Testing Scenarios

The model used in this study was built in the SimaPro software. The method of using existing software has been adopted by researchers such as Farina et al. [29] and Zhang et al. [28].

As the selected impact assessment tool in this research, the ReCiPe Mid-Point (Europe H) method provided an exhaustive evaluation of the environmental implications. This method is capable of generating a total of eighteen diverse impact categories, out of which twelve were considered for the midpoint impact categories analysis, including cli-

mate change, ozone depletion, terrestrial acidification, human toxicity, photochemical oxidant formation, particulate matter formation, terrestrial ecotoxicity, freshwater ecotoxicity, marine ecotoxicity, water depletion, metal depletion, and fossil fuel depletion.

This study analyzed and compared the environmental impact data of brown coal fly ash brick with PC concrete blocks. Two brown coal brick mix designs and a PC mix designs were adopted [33]: (1) the Loy Yang brown coal fly ash (LYFA) mix achieved a 21.7 MPa compressive strength at 28 days, which corresponds to the application as fire bricks in Australia; (2) the Yallourn brown coal fly ash (YFA) mix had a lower compressive strength, 6.8 MPa, and hence, can be used as a general-purpose brick [33]. Table 1 shows the mix proportions of the two types of bricks.

**Table 1.** Geopolymer and PC brick mix design [33].

| Bricks | Mix Design (kg/m³) | | | | | | | | Water to Solid | 28-Day Compressive Strength (MPa) |
|---|---|---|---|---|---|---|---|---|---|---|
| | PC | Brown Coal Ash | Sand | Dust | White Stone (7 mm) | Water | Activator Solution | | | |
| | | | | | | | $Na_2SiO_3$ | NaOH | | |
| PC | 160 | - | 728 | 182 | 291 | 76.8 | - | - | 0.48 | 15.0 |
| LYFA | - | 160 | 728 | 182 | 291 | 10 | 208 | 12 | 0.52 | 21.7 |
| YFA | - | 152 | 689 | 172 | 276 | 0 | 271 | 17 | 0.58 | 6.8 |

### 3. Life Cycle Inventory Analysis

*3.1. Life Cycle Phases*

The LCA phases selected for each brick category are summarized in Table 2. These phases were selected based on the specific production conditions for the geopolymer and PC bricks. The raw material manufacturing stage consisted of the raw material extraction/production phase and the collection/drying process. The brick production phase consisted of both the mixing and heat-curing processes stated in Table 2.

**Table 2.** Detailed life cycle phases considered for the analysis.

| Material | PC | Brown Coal Ash | Fine Aggregate | Coarse Aggregate | NaOH | Sodium Silicate | Geopolymer Brick | PC Brick |
|---|---|---|---|---|---|---|---|---|
| Raw material extraction and production | √ | - | √ | √ | √ | √ | √ | √ |
| Collection and drying | - | √ | - | - | - | - | - | - |
| Transportation | √ | √ | √ | √ | √ | √ | √ | √ |
| Mixing | - | - | - | - | - | - | √ | √ |
| Heat curing | - | - | - | - | - | - | √ | - |
| Distribution | - | - | - | - | - | - | √ | √ |
| Usage | - | - | - | - | - | - | √ | √ |
| End of life | - | - | - | - | - | - | √ | √ |

*3.2. Raw Material Acquisition*

The brown coal ash was obtained from two power plants in the La Trobe Valley, Victoria, Australia, namely Loy Yang (LYFA) and Yallourn (YFA) power plants. Although both power plants are close in proximity, the two ash types vary in composition and properties. The variations in the materials are attributed to differences between the two coal seams and storage regimes. The Loy Yang ash is relatively high in aluminosilicates, while the Yallourn ash is relatively low [34]. The mix design utilized identical specific fine and coarse aggregates. The fine aggregates employed were Chelvon sand and Hanson dust, while the coarse aggregates were Chelvon white stone (7 mm). A Grade-D sodium silicate

solution and a sodium hydroxide (NaOH) 15 M solution were used as the alkali activator for the geopolymer bricks. General-purpose PC was adopted for the PC bricks' production for the comparative study.

### 3.3. Transportation Details

The transportation scenarios and distances for all the types of bricks are summarized in Table 3. The brick manufacturing was considered to be located in Melbourne, Australia. The transportation mode for all phases (including the transportation of raw materials and distribution of bricks) was considered as by road, with diesel heavy trucks. All the travel distance of the transportation of materials was based on real-life data.

**Table 3.** Transportation distances of this LCA study.

| Transportation Stage | Distance (km) |
|---|---|
| LYFA to manufacturing plant | 168 |
| YFA to manufacturing plant | 145 |
| PC to manufacturing plant | 50 |
| Sodium silicate to manufacturing plant | 38.5 |
| NaOH to manufacturing plant | 26.1 |
| Chelvon sand to manufacturing plant | 16.7 |
| Chelvon dust to manufacturing plant | 16.7 |
| White stone to manufacturing plant | 29.8 |
| Distribution | 50 |
| Disposal (landfilling) distance | 56 |

### 3.4. Energy Consumption

The Australian electricity grid mix [35] was used for all processes shown in the system boundary. The primary source of electricity was coal (61%), followed by natural gas (19%), oil (2%), hydropower (7%), wind (6%), solar (3%), bio-energy (2%), and other renewable energy.

## 4. Results

### 4.1. Comparative Analysis

A comparative analysis of the twelve environmental impact categories included in the ReCiPe midpoint methodology was undertaken for the two brown coal geopolymer bricks and compared with the PC bricks. The characterized impact intensities for the twelve midpoint categories are presented in Table 4. All results are presented as the unit of functional performance (compressive strength) for all categories. Figure 2 illustrates the percentage variation for the three brick types for all midpoint categories. The results showed that LYFA had a similar variation for climate change ($1.97 \times 10^1$ kg $CO_2$ eq/m$^3$. MPa) during the "cradle-to-grave" phases compared to the PC bricks ($1.94 \times 10^1$ kg $CO_2$ eq/m$^3$. MPa). However, slightly higher impacts for all other impact categories were observed for LYFA when compared with the PC bricks, except ozone depletion, water depletion, and metal depletion. When considering the "cradle-to-grave" approach, ozone depletion (~27%), water depletion (~30%), and metal depletion (~47%) for LAFA showed reduced environmental impacts compared to the PC bricks. The LAFA geopolymer bricks showed higher impact values for terrestrial acidification (~67%), human toxicity (~40%), photochemical oxidant formation (~34%), particulate matter formation (~51%), terrestrial ecotoxicity (~24%), freshwater ecotoxicity (~94%), marine ecotoxicity (~92%), and fossil fuel depletion (~55%) compared to the PC bricks, as shown in Figure 2. The YFA bricks showed higher impacts ranging between 72% and 76% and 61% and 98% for all midpoint categories compared to the LYFA and PC bricks, respectively.

**Table 4.** Quantified environmental impact values for geopolymer and PC bricks.

| Impact Category | Unit | Impact Intensity | | |
| --- | --- | --- | --- | --- |
| | | **LYFA** | **YFA** | **PC** |
| Climate change | kg $CO_2$ eq/$m^3$. MPa | $1.97 \times 10^1$ | $7.86 \times 10^1$ | $1.94 \times 10^1$ |
| Ozone depletion | kg CFC-11 eq/$m^3$. MPa | $4.53 \times 10^{-7}$ | $1.71 \times 10^{-6}$ | $6.19 \times 10^{-7}$ |
| Terrestrial acidification | kg $SO_2$ eq/$m^3$. MPa | $1.34 \times 10^{-1}$ | $5.58 \times 10^{-1}$ | $4.47 \times 10^{-2}$ |
| Human toxicity | kg 1,4-DB eq/$m^3$. MPa | $3.17 \times 10^0$ | $1.26 \times 10^1$ | $1.91 \times 10^0$ |
| Photochemical oxidant formation | kg NMVOC/$m^3$. MPa | $7.36 \times 10^{-2}$ | $2.97 \times 10^{-1}$ | $4.83 \times 10^{-2}$ |
| Particulate matter formation | kg PM10 eq/$m^3$. MPa | $3.40 \times 10^{-2}$ | $1.39 \times 10^{-1}$ | $1.68 \times 10^{-2}$ |
| Terrestrial ecotoxicity | kg 1,4-DB eq/$m^3$. MPa | $3.50 \times 10^{-4}$ | $1.41 \times 10^{-3}$ | $2.65 \times 10^{-4}$ |
| Freshwater ecotoxicity | kg 1,4-DB eq/$m^3$. MPa | $4.52 \times 10^{-1}$ | $1.92 \times 10^0$ | $2.91 \times 10^{-2}$ |
| Marine ecotoxicity | kg 1,4-DB eq/$m^3$. MPa | $3.91 \times 10^{-1}$ | $1.66 \times 10^0$ | $2.86 \times 10^{-2}$ |
| Water depletion | $m^3$/$m^3$. MPa | $1.46 \times 10^{-1}$ | $5.31 \times 10^{-1}$ | $2.08 \times 10^{-1}$ |
| Metal depletion | kg Fe eq/$m^3$. MPa | $1.16 \times 10^{-1}$ | $4.16 \times 10^{-1}$ | $2.17 \times 10^{-1}$ |
| Fossil fuel depletion | kg oil eq/$m^3$. MPa | $5.61 \times 10^0$ | $2.31 \times 10^1$ | $2.53 \times 10^0$ |

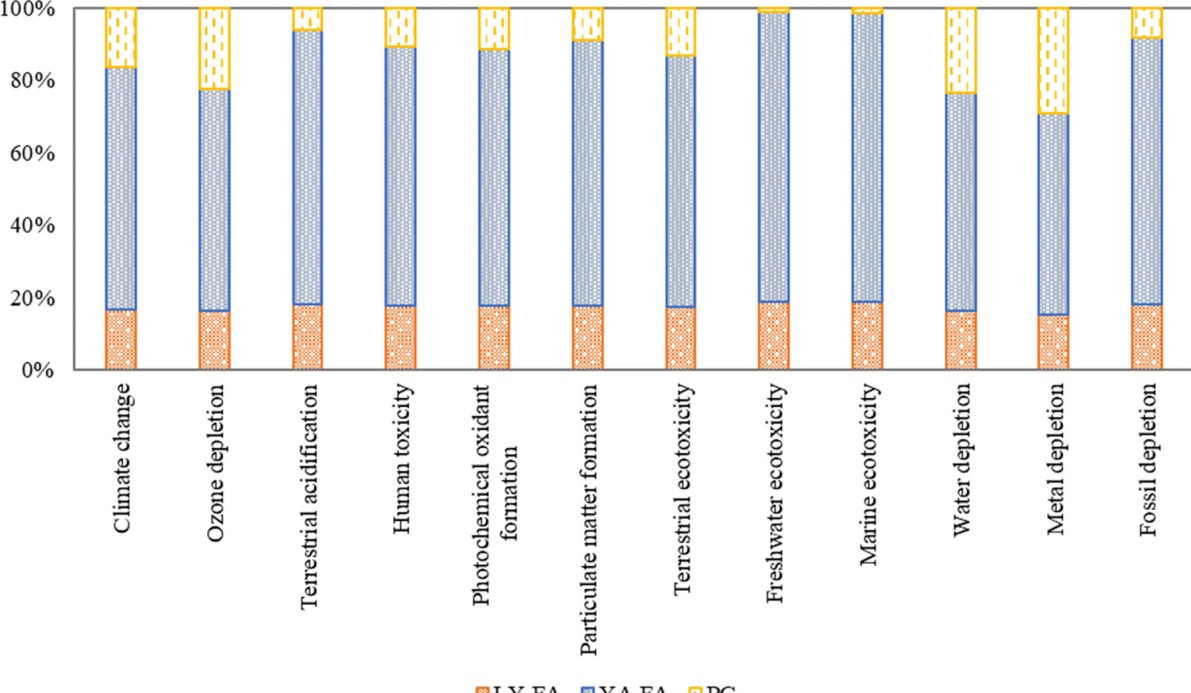

**Figure 2.** Comparative percentage midpoint characterization values for LYFA, YFA, and PC bricks.

*4.2. Contribution Analysis*

The impact categories' proportional impacts and intensities relevant to every life cycle stage of the brick production for the LYFA, YFA, and PC bricks are illustrated with regard to the twelve midpoint categories in Figure 3. The fly ash collection location (transportation distance), mix design, and brick compressive strength were the main differences between the LYFA and YFA bricks. Both the LYFA bricks and YFA bricks displayed a similar percentage variation for all midpoint categories, as shown in Figure 3. Climate change was the highest impact associated with the material manufacturing phase for all types of bricks. However, the total PC brick contributions were more than 80% in the climate change category, while both brown coal geopolymer bricks contributed approximately 62% to climate change in the stage of material manufacturing. Furthermore, fossil fuel

depletion contributed approximately 22% and 9% for the total impact in the stage of material manufacturing for the geopolymer bricks and PC bricks, respectively. The transportation of raw materials, the phase of distribution usage, and the stage of end-of-life showed a similar variation of the percentage for all the bricks. Moreover, the brick manufacturing phase alone accounted for a higher share of climate change (~74%), fossil fuel depletion (~19%), and human toxicity (~2%) for the PC bricks.

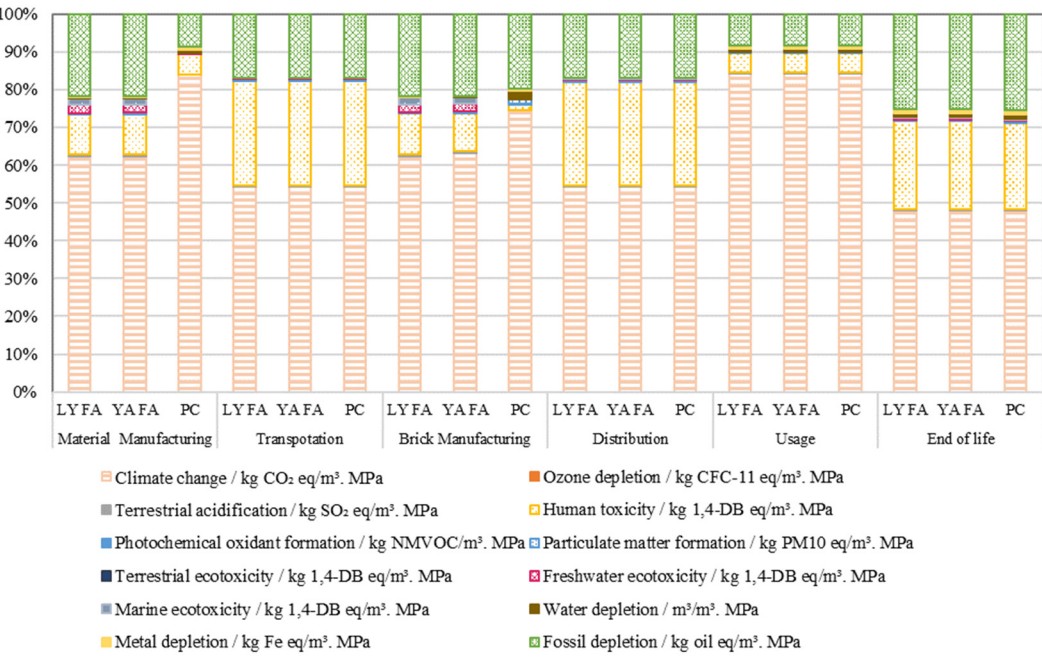

**Figure 3.** Comparison of the percentage of the environmental impacts of brown coal geopolymer bricks and PC bricks.

The impact intensities for the twelve selected environmental impact categories and key LCA phases for the geopolymer and PC bricks are illustrated in Figure 4. The results illustrated that the YFA bricks produced higher impact intensities for all impact categories based on six LCA phases, as verified by the total percentage of the impacts in the comparative analysis. However, the material manufacturing and usage phases showed higher values for both geopolymer and PC bricks in the climate change impact category. A higher percentage for the material manufacturing, usage, and end-of-life phases are reported for ozone depletion (Figure 4b), while the material manufacturing phase was accountable for the highest proportion for LYFA and YFA compared to PC, which were terrestrial acidification (Figure 4c), human toxicity (Figure 4d), photochemical oxidant formation (Figure 4e), particulate matter formation (Figure 4f), terrestrial ecotoxicity (Figure 4g), freshwater ecotoxicity (Figure 4h), and marine ecotoxicity (Figure 4i). Furthermore, material manufacturing alone was responsible for more than a 50% share of the whole process for all impact categories, except ozone depletion and metal depletion, in both geopolymer bricks.

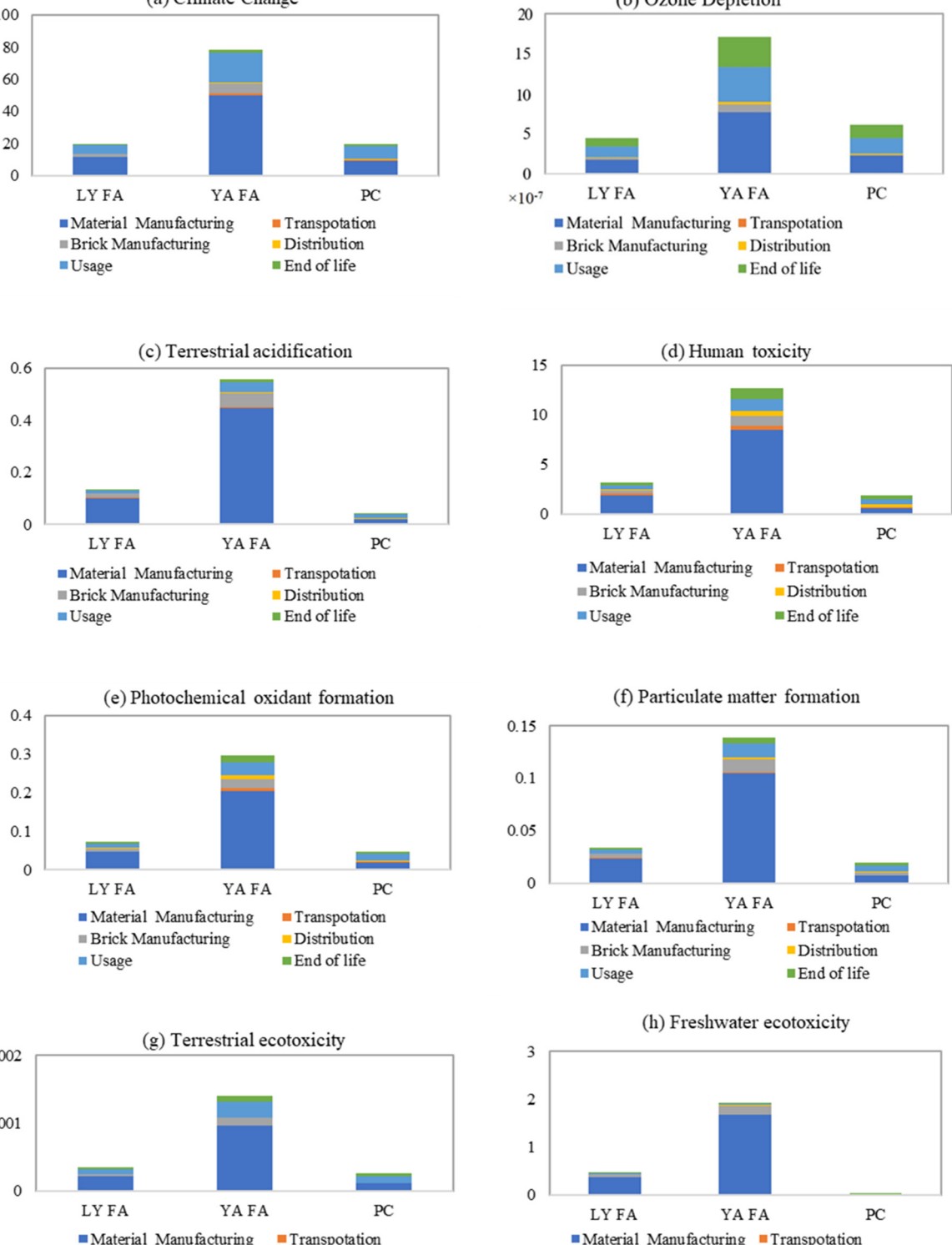

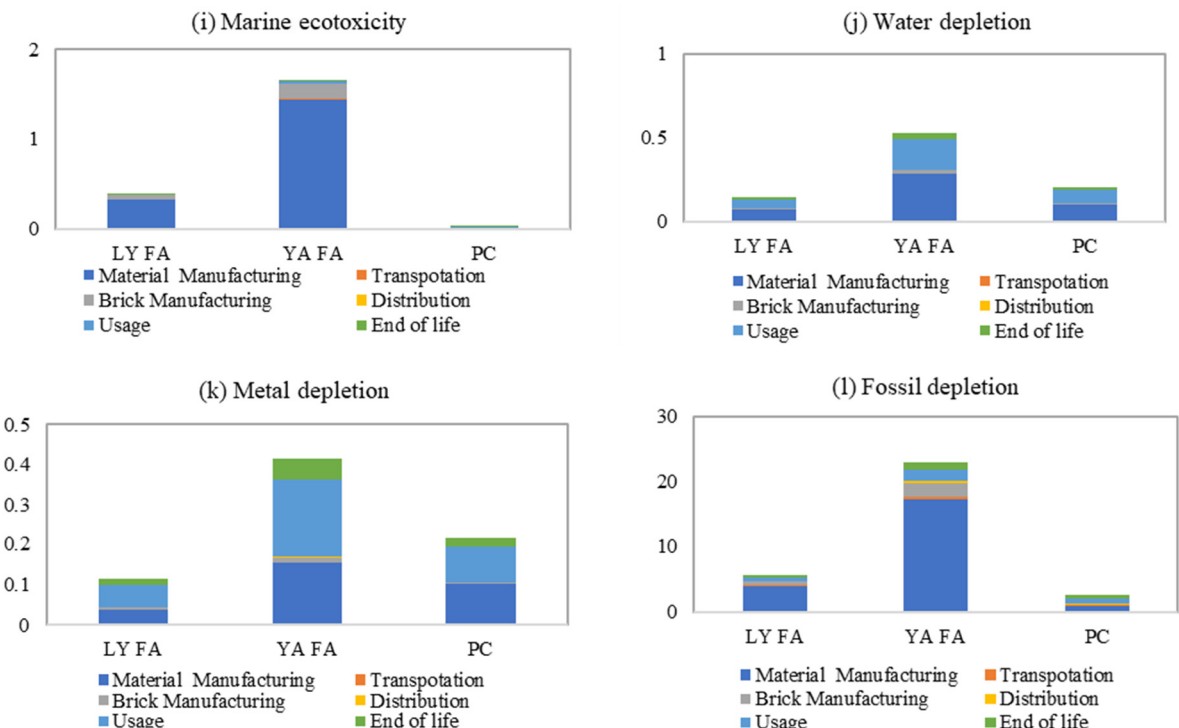

**Figure 4.** Comparison of the environmental impact intensities of brown coal geopolymer bricks and PC bricks; (**a**) climate change (kg $CO_2$ eq/m³. MPa), (**b**) ozone depletion (kg CFC11 eq/m³. MPa), (**c**) terrestrial acidification (kg $SO_2$ eq/m³. MPa), (**d**) human toxicity (kg 1,4-DB eq/m³. MPa), (**e**) photochemical oxidant formation (kg NMVOC/m³. MPa), (**f**) particulate matter formation (kg $PM_{10}$ eq/m³. MPa), (**g**) terrestrial ecotoxicity (kg 1,4-DB eq/m³. MPa), (**h**) freshwater ecotoxicity (kg 1,4-DB eq/m³. MPa), (**i**) marine ecotoxicity (kg 1,4-DB eq/m³. MPa), (**j**) water depletion (m³/m³. MPa), (**k**) metal depletion (kg Fe eq/m³. MPa), and (**l**) fossil fuel depletion (kg oil eq/m³. MPa).

The contribution analysis showed that the material manufacturing phase contributed from 34% to 84%, 32 to 87%, and 32% to 49% of all the impacts for the LYFA, YFA, and PC bricks, respectively. The transportation and distribution phase had the lowest proportion (i.e., 1% to 13%) for all impact categories for all brick types. The brick manufacturing phase contributed 3% to 13% for the LYFA geopolymer bricks, 3% to 10% for the YFA geopolymer bricks, and 1% to 15% for the PC bricks for all impact categories. The usage phase was the highest contributor to the metal depletion impact category for both the geopolymer and PC bricks. The LYFA, YFA, and PC bricks contributed 50%, 47%, and 40% for metal depletion in the usage phase, respectively. The end-of-life phase contributed 1% to 27% of the impact for all impact categories for the geopolymer and PC brick types.

The material manufacturing phase consisted of the material preparation, including the collection, drying, and processing of the raw materials for the geopolymer and PC bricks. Figure 5 shows the detailed comparison of the environmental impacts for each raw material used for the brick production at the material manufacturing stage. The results clearly identified that the alkaline activators were responsible for over 80% of the total impacts associated in the material manufacturing stage for both the LYFA and YFA bricks for all impact categories other than water and metal depletion. For the PC bricks, PC alone was responsible for the highest share among all the impacts during the stage of material manufacturing, other than the water depletion impact category.

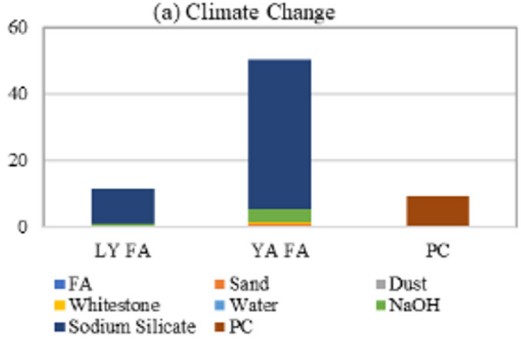

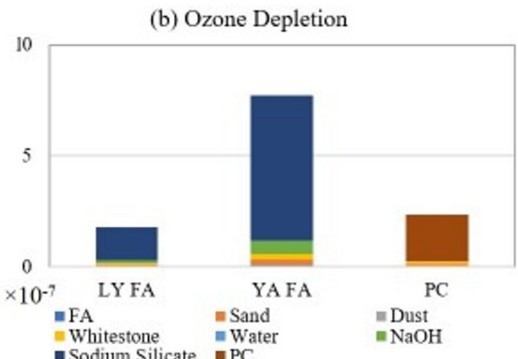

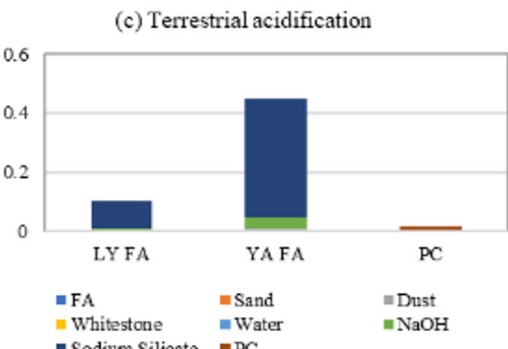

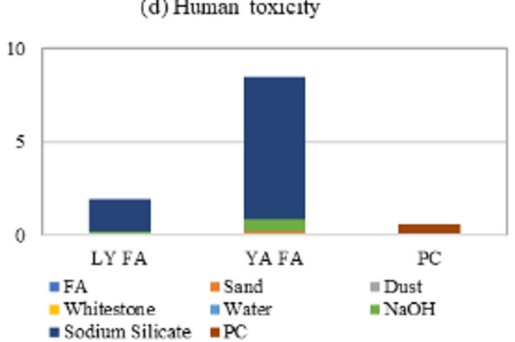

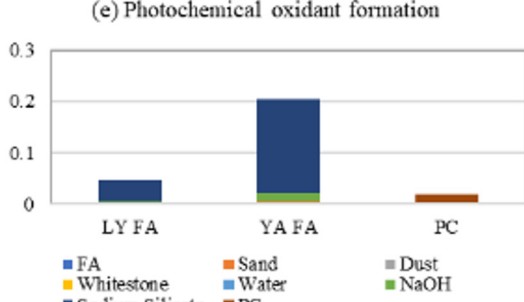

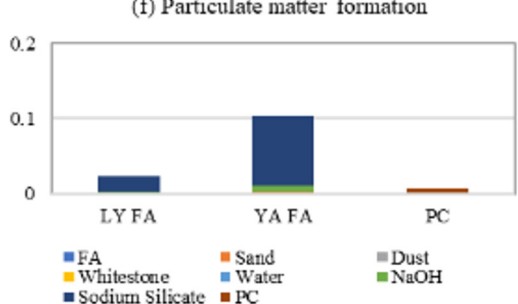

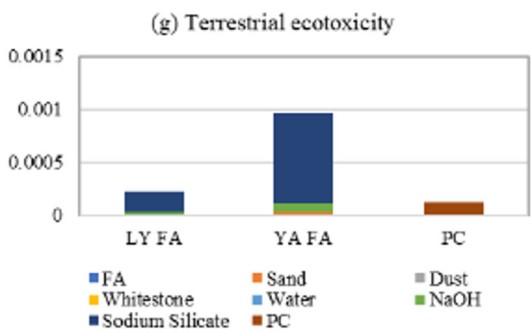

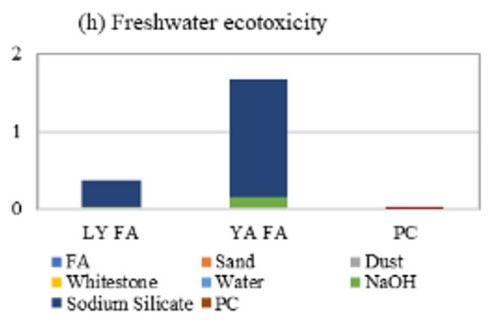

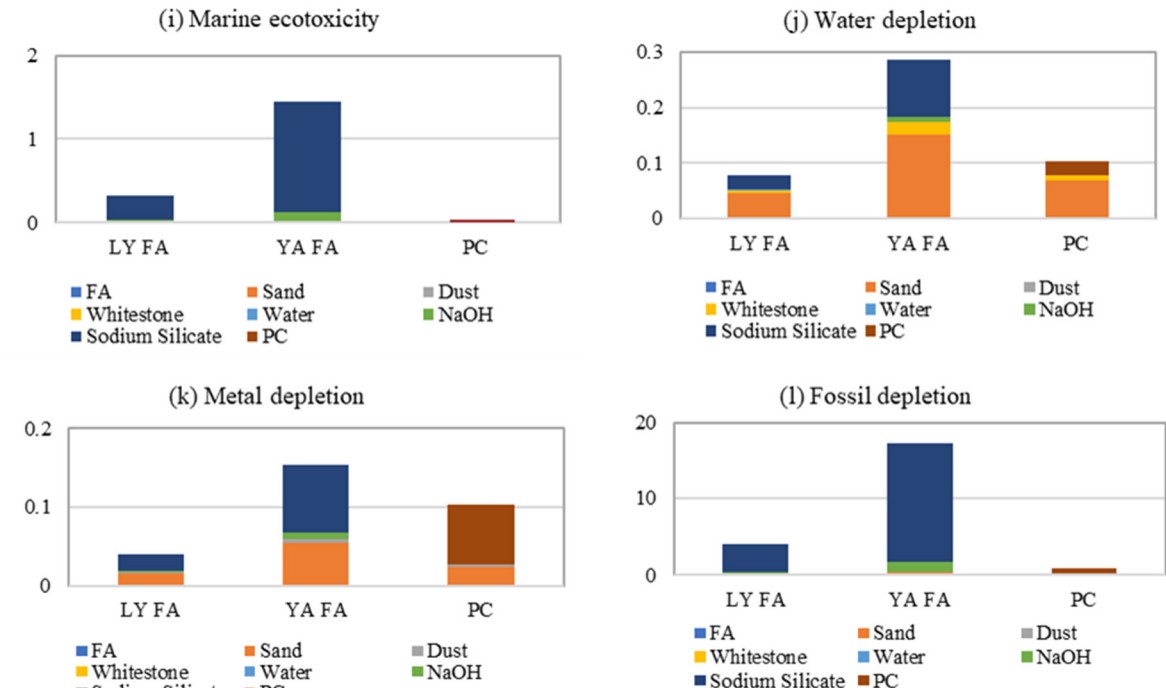

**Figure 5.** Comparison of the environmental impact intensities for raw material manufacturing stage; (**a**) climate change (kg $CO_2$ eq/m³. MPa), (**b**) ozone depletion (kg CFC-11 eq/m³. MPa), (**c**) terrestrial acidification (kg $SO_2$ eq/m³. MPa), (**d**) human toxicity (kg 1,4-DB eq/m³. MPa), (**e**) photochemical oxidant formation (kg NMVOC/m³. MPa), (**f**) particulate matter formation (kg PM10 eq/m³. MPa), (**g**) terrestrial ecotoxicity (kg 1,4-DB eq/m³. MPa), (**h**) freshwater ecotoxicity (kg 1,4-DB eq/m³. MPa), (**i**) marine ecotoxicity (kg 1,4-DB eq/m³. MPa), (**j**) water depletion (m³/m³. MPa), (**k**) metal depletion (kg Fe eq/m³. MPa), and (**l**) fossil fuel depletion (kg oil eq/m³. MPa).

Table 5 summarizes the impact intensities for the brick manufacturing phase. According to the results, the higher impacts associated with the geopolymer brick production phase were due to heat curing in the brick production process. The mixing process was only required in the PC bricks' production phase, while heat curing was not required. When considering the mixing process, the energy consumption for a unit volume was the same for all the brick types. However, the impact intensity was lower for LYFA when compared to the PC brick mixing process, while the YFA brick manufacturing phase showed higher impact intensities for all categories due to lower mechanical performance. The brick manufacturing phase was directly correlated with energy consumption. Fossil fuel depletion and climate change were identified as the categories with the highest impact during the brick production stage.

**Table 5.** Detailed impact category intensities of the brick production phase.

| Impact Category | Unit | Mixing | | | Heat Curing | | |
|---|---|---|---|---|---|---|---|
| | | **LYFA** | **YFA** | **PC** | **LYFA** | **YFA** | **PC** |
| Climate change | kg $CO_2$ eq/m³. MPa | $3.83 \times 10^{-2}$ | $4.18 \times 10^{-1}$ | $1.90 \times 10^{-1}$ | $1.70 \times 10^{0}$ | $5.76 \times 10^{0}$ | $0.00 \times 10^{0}$ |
| Ozone depletion | kg CFC-11 eq/m³. MPa | $5.63 \times 10^{-10}$ | $1.01 \times 10^{-8}$ | $4.58 \times 10^{-9}$ | $2.50 \times 10^{-8}$ | $8.46 \times 10^{-8}$ | $0.00 \times 10^{0}$ |
| Terrestrial acidification | kg $SO_2$ eq/m³. MPa | $3.44 \times 10^{-4}$ | $7.74 \times 10^{-4}$ | $3.51 \times 10^{-4}$ | $1.53 \times 10^{-2}$ | $5.17 \times 10^{-2}$ | $0.00 \times 10^{0}$ |
| Human toxicity | kg 1,4-DB eq/m³. MPa | $6.51 \times 10^{-3}$ | $8.17 \times 10^{-3}$ | $3.71 \times 10^{-3}$ | $2.89 \times 10^{-1}$ | $9.78 \times 10^{-1}$ | $0.00 \times 10^{0}$ |
| Photochemical oxidant formation | kg NMVOC/m³. MPa | $1.56 \times 10^{-4}$ | $1.61 \times 10^{-3}$ | $7.29 \times 10^{-4}$ | $6.93 \times 10^{-3}$ | $2.34 \times 10^{-2}$ | $0.00 \times 10^{0}$ |
| Particulate matter formation | kg PM10 eq/m³. MPa | $7.99 \times 10^{-5}$ | $4.19 \times 10^{-4}$ | $2.85 \times 10^{-3}$ | $3.55 \times 10^{-3}$ | $1.20 \times 10^{-2}$ | $0.00 \times 10^{0}$ |
| Terrestrial ecotoxicity | kg 1,4-DB eq/m³. MPa | $7.37 \times 10^{-7}$ | $4.99 \times 10^{-6}$ | $2.26 \times 10^{-6}$ | $3.27 \times 10^{-5}$ | $1.11 \times 10^{-4}$ | $0.00 \times 10^{0}$ |

| | | | | | | | |
|---|---|---|---|---|---|---|---|
| Freshwater ecotoxicity | kg 1,4-DB eq/m³. MPa | $1.30 \times 10^{-3}$ | $2.12 \times 10^{-4}$ | $9.59 \times 10^{-5}$ | $5.78 \times 10^{-2}$ | $1.95 \times 10^{-1}$ | $0.00 \times 10^{0}$ |
| Marine ecotoxicity | kg 1,4-DB eq/m³. MPa | $1.12 \times 10^{-3}$ | $2.35 \times 10^{-4}$ | $1.06 \times 10^{-4}$ | $4.99 \times 10^{-2}$ | $1.69 \times 10^{-1}$ | $0.00 \times 10^{0}$ |
| Water depletion | m³/m³. MPa | $8.77 \times 10^{-5}$ | $1.26 \times 10^{-2}$ | $5.70 \times 10^{-3}$ | $3.90 \times 10^{-3}$ | $1.32 \times 10^{-2}$ | $0.00 \times 10^{0}$ |
| Metal depletion | kg Fe eq/m³. MPa | $7.36 \times 10^{-5}$ | $3.36 \times 10^{-3}$ | $1.52 \times 10^{-3}$ | $3.27 \times 10^{-3}$ | $1.11 \times 10^{-2}$ | $0.00 \times 10^{0}$ |
| Fossil fuel depletion | kg oil eq/m³. MPa | $1.34 \times 10^{-2}$ | $1.10 \times 10^{-1}$ | $4.99 \times 10^{-2}$ | $5.94 \times 10^{-1}$ | $2.01 \times 10^{0}$ | $0.00 \times 10^{0}$ |

### 4.3. Benefit Analysis

The location in which the fly ash was obtained led to different environmental impacts of the brown coal fly ash in the bricks' production. Figure 6 illustrates the environmental benefits obtained by utilizing the brown coal ash from landfills. A substantial decrease in the category of human toxicity is noted, corresponding to LYFA being 3.18 kg 1,4-DB eq/m³. MPa and YFA being 9.63 kg 1,4-DB eq/m³. MPa. There were decreased impacts from freshwater ecotoxicity of 0.27 1,4-DB eq/m3 MPa and marine water ecotoxicity of 0.82 1,4-DB eq/m³ MPa for LYFA and $2.56 \times 10^{-1}$ 1,4-DB eq/m³ MPa and $7.75 \times 10^{-1}$ 1,4-DB eq/m³ MPa for YFA, respectively. Climate change, acidification, photochemical oxidant formation, particulate matter formation, and fossil fuel depletion demonstrated benefits of less than 1% for LYFA and YFA. The highest benefit was obtained in the category of human toxicity, 100.34% and 76.19 % for LYFA and YFA, respectively. Furthermore, terrestrial, water, and marine ecotoxicity accounted for the second-largest benefits for both LYFA and YFA due to the avoidance of the storage of brown coal ash.

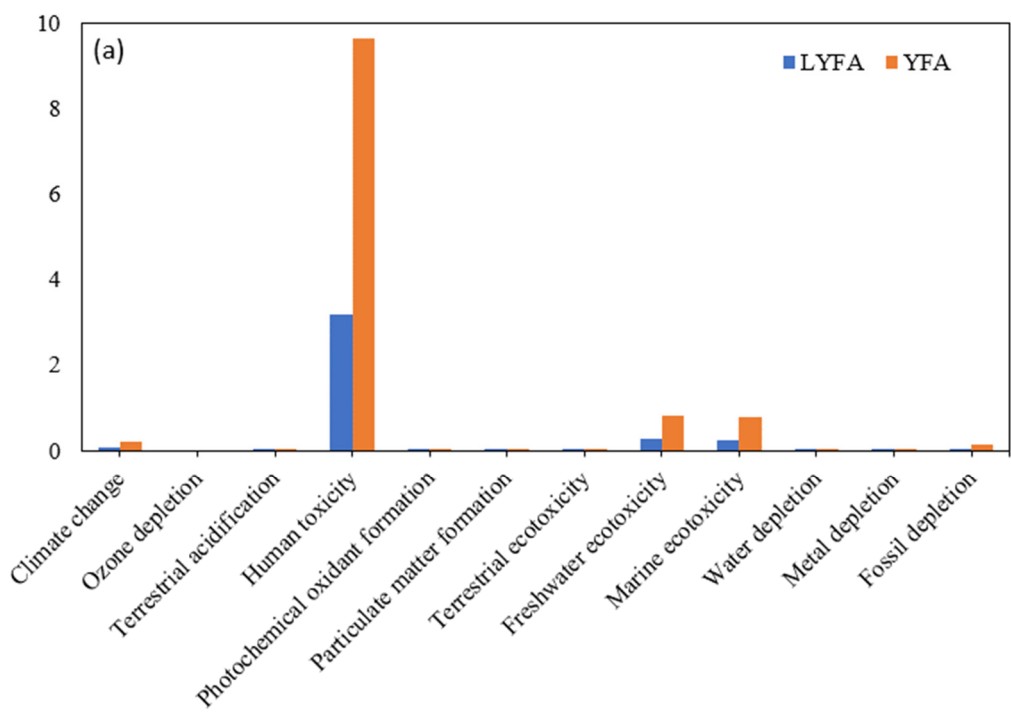

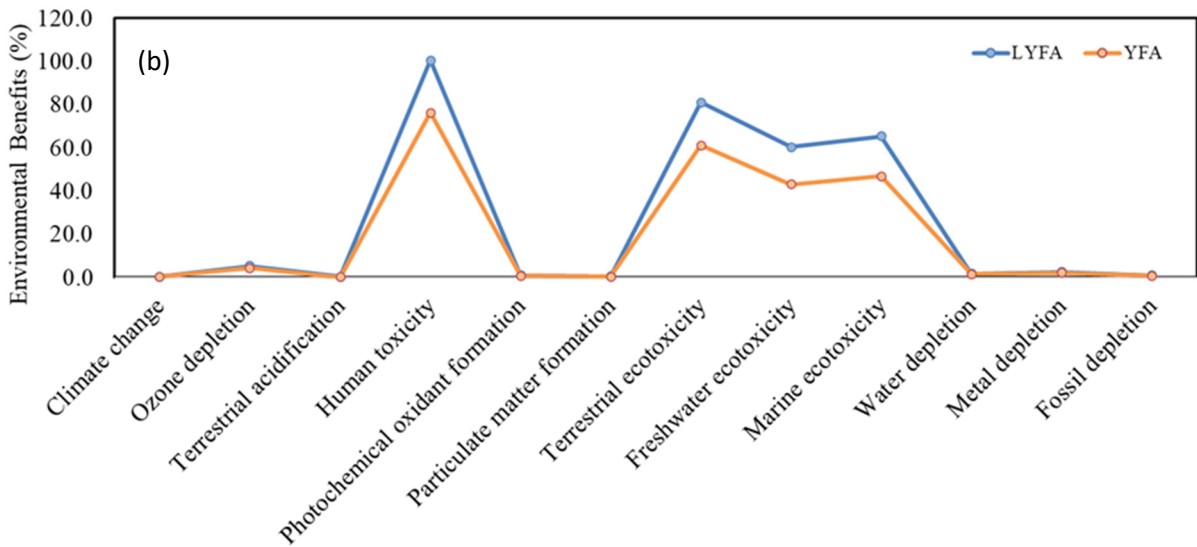

**Figure 6.** Utilization of waste brown coal ash's benefit for environmental (**a**) intensities and (**b**) variation of the percentage.

### 4.4. Cost Analysis

The life cycle cost analysis encompassed the extraction of the raw materials, the transportation of the raw materials, and the bricks' manufacturing (cradle-to-gate). The outputs are summarized in Table 6. Figure 7 compares the percentage distribution of the brown coal ash bricks and PC bricks for the raw material manufacturing and transportation phases. The unit costs for the extraction/production and transportation of raw materials were determined by accounting for current market values and sourcing from local Australian suppliers. PC had a cost of AUD 64, while brown coal ash can be freely obtained. When considering both LYFA and YFA geopolymer bricks, sodium silicate at the stage of material manufacture was the largest contributor to the cost, 51% and 57%, respectively. Sand had the highest cost attribution for the PC bricks, which constituted nearly 52% of the total cost.

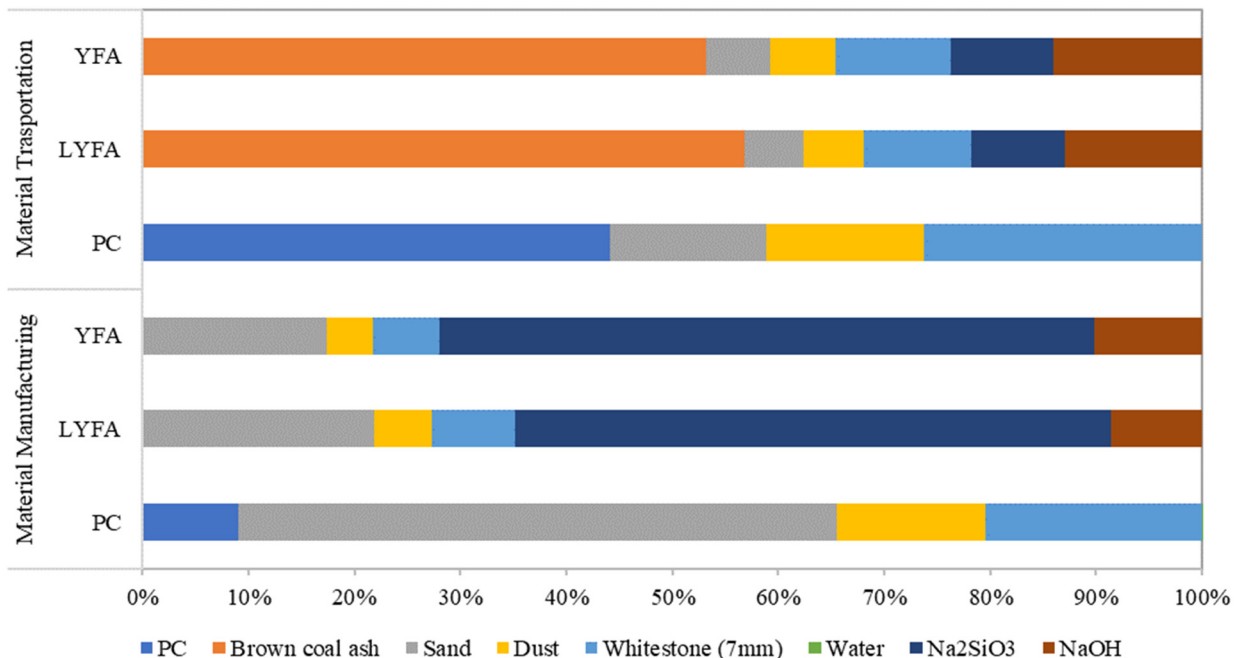

**Figure 7.** Percentage cost distribution for raw material production and transportation stages for LYFA, YFA, and PC bricks.

**Table 6.** Cost analysis of the "cradle-to-gate" for geopolymer and PC bricks.

| Phases | Brick | Cost (AUD) per 1 m³ | | | | | | | | |
|---|---|---|---|---|---|---|---|---|---|---|
| | | PC | Brown Coal Ash | Sand | Dust | White Stone (7 mm) | Activator Solution | | Total Cost per 1 m³ | Cost per Brick |
| | | | | | | | Na₂SiO₃ | NaOH | | |
| Material Manufacturing | PC | 64.00 | 0.00 | 400.40 | 100.10 | 144.77 | 0.00 | 0.00 | 709.28 | 1.33 |
| | LYFA | 0.00 | 0.00 | 400.40 | 100.10 | 144.77 | 1033.11 | 157.24 | 1835.63 | 3.45 |
| | YFA | 0.00 | 0.00 | 378.95 | 94.60 | 137.31 | 1346.03 | 222.76 | 2179.65 | 4.10 |
| Material Transportation | PC | 17.30 | 0.00 | 5.78 | 5.81 | 10.31 | 0.00 | 0.00 | 39.21 | 0.07 |
| | LYFA | 0.00 | 58.14 | 5.78 | 5.81 | 10.31 | 9.03 | 13.32 | 102.40 | 0.19 |
| | YFA | 0.00 | 50.18 | 5.78 | 5.81 | 10.31 | 9.03 | 13.32 | 94.44 | 0.18 |
| Brick Manufacturing | PC | | | | | - | | | 10.80 | 0.02 |
| | LYFA | | | | | | | | 69.40 | 0.13 |
| | YFA | | | | | | | | 69.40 | 0.13 |

LYFA contributed the most to the material transport stage, followed by YFA and PC. When compared to PC, LYFA and YFA had the largest cost of transportation (i.e., AUD 58.14 and AUD 50.18). In the production of the PC bricks, the transportation of PC cost AUD 17.3, making it the highest contributor in the transportation stage. Hence, when consideration was given to the overall transport costs, the geopolymer bricks were more costly compared to the PC bricks.

Only the electricity consumption during the brick manufacturing stage was regarded as a part of the brick manufacturing process. The high costs involved in the brick manufacturing stage of the brown coal bricks were attributed to the process of heat curing. The heat curing demanded a large amount of electricity and, therefore, resulted in higher costs. However, for the PC and brown coal geopolymer bricks, only about 1.4% and 3.4% of the total cost needed to be accounted for in the material manufacturing stage.

The PC concrete bricks' cost for cradle-to-gate manufacturing was AUD 1.43, while the LYFA and YFA geopolymer bricks cost AUD 3.78 and AUD 4.41, respectively. Both geopolymer bricks had a total brick cost increment of 162% (LYFA) and 167% (YFA) as

compared to the PC bricks. Furthermore, the YFA bricks had a 14% higher total cost compared to the LYFA bricks.

## 5. Discussion

The comparative analysis of the two brown coal geopolymer bricks with traditional PC bricks highlighted that the YFA bricks had the highest associated impacts, followed by the LYFA bricks, both of which were higher than the traditional PC bricks. The main reason for the higher impacts associated with the YFA geopolymer bricks was the lower compressive strength. The LYFA and YFA bricks had a slight variation in the impact values due to the minor differences in the transportation phase of the bricks' production. Moreover, during the "cradle-to-grave" phases, LYFA demonstrated slightly higher climate change variation compared with the PC bricks. This was due to the higher impact in the manufacturing phase for the LYFA bricks, while both the manufacturing and usage phases contributed to the higher climate change impact for the PC bricks. Although a waste by-product was utilized for the geopolymer bricks' production, a higher impact was observed due to the alkali activators. From the detailed analysis of the LCA, it was clearly noted that the alkaline activators contributed significantly to the climate change impact category. Furthermore, LYFA and YFA not only had a greater impact on climate change, but also on all the other impact categories (except water depletion) during the material manufacturing stage due to the use of the alkaline activators (i.e., sodium silicate and sodium hydroxide). The sodium silicate production process was responsible for the higher impact of all other impact categories, mainly as a result of calcination in the manufacturing process. The manufacturing of sodium silicate involves dissolution, processing, and filtration, which entails a significant energy consumption (i.e., electricity and heat) and yields significant air and water emissions, as well as solid waste [36,37]. Additionally, the electrolysis of sodium chloride is an energy-intensive process in the manufacture of NaOH. This results in higher emissions and environmental impact due to electricity use, natural gas use, and waste disposal [38].

According to the detailed contribution analysis of the material manufacturing phase, PC contributed the greatest share for all impact categories (except water depletion). Clinker production during the PC manufacturing process is an energy-intensive activity that is responsible for the highest emissions and environmental impacts [39]. The brick production phase alone contributed a minor impact for all brick types. Hence, the higher impacts for the LYFA and YFA bricks than the PC bricks were primarily due to the heat-curing process, which consisted of additional energy consumption during the brick manufacturing process. In Australia, the national energy grid includes non-renewable energy, but is primarily coal combustion (almost 60% of the total energy grid). Hence, direct emissions derived from coal combustion have a primary effect on energy consumption during activator production and PC production.

The remaining categories, i.e., terrestrial acidification, human toxicity, photochemical oxidant formation, particulate matter formation, terrestrial ecotoxicity, freshwater ecotoxicity, marine ecotoxicity, and fossil fuel depletion, showed higher impacts for LYFA compared to the PC bricks, even with relatively higher compressive strength. This was again directly related to the higher energy consumption during the alkaline activator production. However, for the YFA bricks, all these impacts had significantly higher values due to their lower compressive strength.

PC production is the principal process responsible for metal depletion. Here, the term metal depletion focuses on the depletion of the resource, except for fossil fuels. This is mainly due to the consumption of natural resources such as limestone and clay and silica stone during clinker manufacturing through the pyrolysis process [40]. The resource depletion that occurred during the PC bricks' production was larger than the resource depletion that occurred for the LYFA geopolymer bricks' production.

Human toxicity was the second-highest impact for the transportation, distribution and end-of-life phases. This was due to the leaching of toxic material from diesel consumption in the transportation of the materials and other products (waste and bricks). The higher human toxicity in the geopolymer bricks was due to the release of toxic elements during the production of sodium silicate.

Water depletion is defined as water scarcity, which means a lack of sufficient available freshwater resources to meet the water demand. According to the results, sand (the extraction of the raw material) was the principal reason for the highest quantity of water depletion, making the material manufacturing stage the phase with the highest impact regarding the water depletion impact category. Generally, irrigation wells and groundwater sources are seriously threatened due to excessive sand extraction near rivers, which negatively affects groundwater recharge. In addition, falling groundwater levels are a major threat to water supplies, exacerbating the occurrence (frequency and periodicity) and severity of droughts, as tributaries of major rivers dry up when sand extraction meets a particular threshold [41]. This was the reason for the higher impact intensities associated with sand in all bricks, including the respective compressive strength within the selected mix design. However, the manufacturing of PC contributed to the water depletion minimally for the PC bricks, while the alkaline activators were the second-highest material resulting in water depletion for the geopolymer bricks. Hence, the impact on water depletion was not significantly affected by the type of brick (geopolymer or PC bricks).

The most-encouraging result was the reduction of ozone depletion by using the brown coal ash in the geopolymer bricks. This was primarily due to the elimination of the PC found in traditional PC bricks. Ozone depletion occurs when the anthropogenic emissions (chlorine and bromine atoms) come into contact with ozone in the stratosphere [42]. This is principally influenced by chlorinated or brominated hydrocarbons emitted during the production of fossil fuels, which was higher in the PC bricks than both geopolymer bricks.

The benefits analysis showed a performance improvement in sustainability due to the reduction of waste disposal (brown coal ash). This was because the waste was transformed into a useful and valuable product. A significant benefit with regard to the impact intensities was shown for human toxicity, fresh water ecotoxicity, and marine water ecotoxicity compared to the impact intensities for the LYFA and YFA bricks. Dumping of fly ash leads to contaminated water and soil with heavy metals and other toxic elements present in the ash itself. Coal ash includes toxic elements, i.e., arsenic (As), barium (Ba), boron (B), beryllium (Be), cadmium (Cd), chromium (Cr), cobalt (Co), lead (Pb), lithium (Li), manganese (Mn), mercury (Hg), molybdenum (Mo), radium (Ra), selenium (Se), thallium (Tl), and other hazardous chemicals [43]. This may be responsible for a range of health problems, affecting every major organ in the body. These effects include cancer, kidney disease, infertility, and compromising the nervous system, especially in children [43]. In addition, brown coal ash pollutes the soil and water sources surrounding coal-fired power stations. Vegetation growing in the vicinity has higher levels of heavy substances (i.e., selenium, zinc, nickel, copper, manganese, cadmium, and lead) from elements leaching from fly ash [44]. Researchers have also discovered that contaminants from coal ash, such as selenium and arsenic, accumulate to "very high concentrations" in fish and wildlife exposed to coal dump leachate or run-off and that these accumulated toxins could eventually lead to deformities or the death of the animal [45]. The use of brown coal ash to produce bricks, therefore, mitigates the environmental impacts of human, water, marine, and soil toxicity.

The life cycle cost analysis illustrated that the stage of raw material manufacturing was responsible for the higher cost associated with the brown coal geopolymer bricks as compared to the PC bricks. The alkaline activator was the main reason for this increased cost. The brick production costs were also higher for the brown coal bricks due to the cost of electricity consumption associated with the thermal curing process. However, in terms

of cost, the transport and brick-making stages were of minor relevance compared to the raw material manufacturing stage.

Currently, coal-fired electricity production is recognized as un-environmentally friendly [46], and many countries are seeking "cleaner" energy, such as renewable energy including solar, hydro, wind, and bio. This will lead to a decrease in coal fly ash supply, which could result in the limited production of brown coal fly ash. However, the high penetration of renewable energy can increase the risk of power outages in the absence of an adequate protective measure [47,48]. Fossil energy is still a reliable energy source and accounted for 75% of the global net electricity generation in 2017 [49]. The evidence shows that coal is still the world's largest single source of electricity, set to still contribute 22% in 2040. In Southeast Asia, coal will provide 39% of electricity in 2040 [50]. In Australia, coal-fired electricity occupies 61% of the electricity production [35]. Since many rely on coal as a crucial electricity production source, the waste by-product, coal fly ash, still needs to be treated. Using coal fly ash in brick production can be a suitable method or this.

### 6. Conclusions and Future Research

Life cycle assessment was employed to carry out an environmental analysis for two brown coal ash geopolymer bricks. The following conclusions can be made based on the results:

- The Loy Yang FA (LYFA) bricks demonstrated slightly higher climate change impact intensities compared to the Portland cement (PC) bricks.
- The Yallourn FA (YFA) bricks showed higher environmental impact intensities for all midpoint categories when compared to both the LYFA and PC bricks due to the lower compressive strength.
- Fossil fuel depletion and climate change were identified as the highest impacted categories during the brick production stage.
- The combination of sodium silicate and sodium hydroxide was responsible for approximately 90% of the total impact for all categories except metal (~50%) and water depletion (~30%) for both brown coal geopolymer bricks.
- Terrestrial acidification, human toxicity, photochemical oxidant formation, particulate matter formation, terrestrial ecotoxicity, freshwater ecotoxicity, marine ecotoxicity, and fossil fuel depletion showed higher impacts for the LYFA bricks compared to the PC bricks.
- Significant environmental benefits in terms of human, freshwater, and marine water ecotoxicity can be obtained by utilizing brown coal ash for the brick manufacturing process
- The most-significant benefits for the LYFA geopolymer bricks over the PC bricks were recorded for the ozone depletion, water depletion, and metal depletion (natural resources other than fossil fuels) categories due to the replacement of PC as a raw material.

The study's findings indicated that there is considerable potential for reducing the environmental impact of the brown coal geopolymer bricks, especially the LYFA bricks, compared to the PC bricks. The replacement of fossil fuels with renewable energy sources during heat curing and the optimization of the activator concentration, type, and ratio can significantly reduce emissions and energy consumption [46,48,51]. Additionally, proper precautions during chemical activator handling and usage can mitigate human toxicity risks. Future research should consider these factors to minimize the environmental impacts during brown coal geopolymer brick production.

Furthermore, energy consumption and material preparation are critical issues that require attention in brick production. Similar results were found in [52,53]. Moving material and brick production to locations where renewable energy sources are available can help control energy consumption. Furthermore, end-of-life management considerations can be included to calculate environmental impacts and benefits.

The study did not consider the durability or lifespan of the geopolymer and PC bricks. However, research has shown that geopolymers' durability can be superior to PC concrete [54–56]. An extended lifespan could enhance the environmental benefits and promote sustainability in future building construction applications.

Fly ash storage is a significant contributor to air, water, and soil pollution, which can be harmful to humans, biodiversity, and flora and fauna. Therefore, ecotoxicology impacts, including terrestrial, freshwater, and marine toxicity, should be considered to improve sustainability in construction. Leaching tests, chemical analyses, and toxicity tests can evaluate their ecotoxicity. Accounting for these impacts can help determine the potential environmental risks associated with the materials used in the construction sector.

**Author Contributions:** Conceptualization, D.W.L. and C.G.; methodology, S.F. and J.Z.; software, J.Z. and S.F.; validation, J.Z. and S.F.; formal analysis, J.Z.; resources, D.W.L. and C.G.; writing—original draft preparation, S.F., J.Z., D.W.L., G.Z. and C.G.; writing—review and editing, D.W.L., G.Z., J.Z. and M.S.; visualization, G.Z. and M.S.; supervision, D.W.L., G.Z., S.S. and C.G.; project administration, D.W.L., S.S., C.G. and G.Z. All authors have read and agreed to the published version of the manuscript.

**Funding:** This research received no external funding.

**Institutional Review Board Statement:** Not applicable.

**Informed Consent Statement:** Not applicable.

**Data Availability Statement:** Data is available on request.

**Conflicts of Interest:** The authors declare no conflict of interest.

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
