# Peer review of "Life Cycle Assessment for Geopolymer Concrete Bricks Using Brown Coal Fly Ash"

_sustainability, doi:10.3390/su15097718_

Round 1

Reviewer 1 Report

The article submitted for review concerns the Life Cycle Assessment for Geopolymer Concrete Bricks using Brown Coal Fly Ash. This study analyzes and compares the environmental impact data of brown coal fly ash brick with PC concrete blocks.

The article was written in a correct and interesting way with an appropriate presentation of the research plan and a summary of the results obtained.

Nevertheless, it is suggested to correct, clarify or supplement the following:

- on page 15, Fig. 6 it is suggested to correct the markings in the drawing. The juxtaposed graphs have similar markings, which is difficult to read;

- why the impact of changes in the thermal insulation of materials (heat conductivity coefficient) on heat losses through penetration, which has a significant impact in the use phase of buildings (heating or cooling), has not been analysed. Please comment on this;

- what are the long-term prospects for obtaining Brown Coal Fly Ash and their availability? In many countries, Fly Ash is becoming a scarce commodity, which was not taken into account at the stage of previous analyses. Has this been reviewed by the authors?

- supplement conclusions with more references to secondary literature.

Author Response

Life Cycle Assessment for Geopolymer Concrete Bricks using Brown Coal Fly Ash

We thank the reviewers for their valuable suggestions to improve the quality of our manuscript. We have incorporated them at appropriate places in the revised manuscript.

Based on the reviewers’ comments, the following changes have been made:

  1. Improved the graph quality
  2. Strengthened the discussion and conclusion, analysis the availability of the brown coal fly ash for brick production. 8 new references have been added:
  • Lozano-Miralles, J.A.; Hermoso-Orzáez, M.J.; Gago-Calderón, A.; Brito, P. LCA Case Study to LED Outdoor Luminaries as a Circular Economy Solution to Local Scale. Sustainability 2020, 12, 190. https://doi.org/10.3390/su12010190
  • Asif, Z., et al., Update on air pollution control strategies for coal-fired power plants. Clean Technologies and Environmental Policy, 2022. 24(8): p. 2329-2347.
  • Liu, D., X. Zhang, and C.K. Tse, Effects of High Level of Penetration of Renewable Energy Sources on Cascading Failure of Modern Power Systems. IEEE Journal on Emerging and Selected Topics in Circuits and Systems, 2022. 12(1): p. 98-106.
  • Yi, W., D.J. Hill, and Y. Song. Impact of High Penetration of Renewable Resources on Power System Transient Stability. in 2019 IEEE Power & Energy Society General Meeting (PESGM). 2019.
  • Wang, Y., et al., Vulnerability of existing and planned coal-fired power plants in Developing Asia to changes in climate and water resources. Energy & Environmental Science, 2019. 12(10): p. 3164-3181.
  • Association, W.C. COAL & ELECTRICITY. 2020 [cited 2023 April 10]; Available from: (https://www.worldcoal.org/coal-facts/coal-electricity/).
  • Marwa, D., et al., A comparative study of life cycle carbon emissions and embodied energy between sun-dried bricks and fired clay bricks. Journal of Cleaner Production, 2020. 275: p. 122998.
  • Ramos Huarachi, D.A., et al., Life cycle assessment of traditional and alternative bricks: A review. Environmental Impact Assessment Review, 2020. 80: p. 106335. 
  1. Added references in conclusion
  2. Improved the language, especially in the discussion section.

Attached file provide a detailed reposne to reviewers comments.

Reviewer 2 Report

Interesting work for the Evaluation of the life cycle of geopolymer concrete bricks using lignite fly ash

The work planted the objectives correctly. The introduction is solidly argued with references to similar works and studies in the line of the work carried out. The LCA methodology applied is clear and scientifically well formulated, supported by real data and the results in the life cycle analysis provide very interesting data for researchers in this field. The discussion of the results in a comparative way provides relevance and technical application of the process.

The results provide scientific novelty showing the environmental advantages of lignite bricks and their potential use in sustainable construction.

We recommend implementing the bibliography a little more with references to similar studies, for example:

As an example:

Lozano-Miralles, J.A.; Hermoso-Orzáez, M.J.; Gago-Calderón, A.; Brito, P. LCA Case Study to LED Outdoor Luminaries as a Circular Economy Solution to Local Scale. Sustainability 2020, 12, 190. https://doi.org/10.3390/su12010190

Author Response

(The authors gave the same response as above.)

Reviewer 3 Report

The manuscript explores the life cycle assessment of geopolymer concrete blocks using lignite fly ash, which has important theoretical value and engineering application value. However, the manuscript needs further improvement before it can be published, with the following suggestions:

1. Since it is an evaluation of the whole life cycle of geopolymer concrete blocks, is it necessary to establish a systematic evaluation model with clear indicators and reasonable weights?

2. As a new building material, it is necessary for the author to analyze the mechanical properties of the new brick to prove that it meets the requirements of the building. At the same time, it is necessary to verify the mechanical properties by some basic mechanical tests.

3. The author should carefully check the structure of the sentences and the coherence of the preceding and following sentences. Also, the sentences in the discussion section need to be simplified.

Author Response

(The authors gave the same response as above.)
